# Acute Coronary Syndrome, Stroke, and Mortality after Community-Acquired Pneumonia: Systematic Review and Meta-Analysis

**DOI:** 10.3390/jcm12072577

**Published:** 2023-03-29

**Authors:** Edinson Dante Meregildo-Rodriguez, Martha Genara Asmat-Rubio, Mayra Janett Rojas-Benites, Gustavo Adolfo Vásquez-Tirado

**Affiliations:** 1Escuela de Medicina, Universidad César Vallejo, Trujillo 13001, Peru; 2Escuela de Posgrado, Universidad Privada Antenor Orrego, Trujillo 13008, Peru; 3Escuela de Medicina, Universidad Privada Antenor Orrego, Trujillo 13008, Peru

**Keywords:** pneumonia, acute coronary syndrome, myocardial infarction, stroke, systematic review

## Abstract

One-third of adult inpatients with community-acquired pneumonia (CAP) develop acute coronary syndrome (ACS), stroke, heart failure (HF), arrhythmias, or die. The evidence linking CAP to cardiovascular disease (CVD) events is contradictory. We aimed to systematically review the role of CAP as a CVD risk factor. We registered the protocol (CRD42022352910) and searched for six databases from inception to 31 December 2022. We included 13 observational studies, 276,109 participants, 18,298 first ACS events, 12,421 first stroke events, 119 arrhythmic events, 75 episodes of new onset or worsening HF, 3379 deaths, and 218 incident CVD events. CAP increased the odds of ACS (OR 3.02; 95% CI 1.88–4.86), stroke (OR 2.88; 95% CI 2.09–3.96), mortality (OR 3.22; 95% CI 2.42–4.27), and all CVD events (OR 3.37; 95% CI 2.51–4.53). Heterogeneity was significant (I^2^ = 97%, *p* < 0.001). Subgroup analysis found differences according to the continent of origin of the study, the follow-up length, and the sample size (I^2^ > 40.0%, *p* < 0.10). CAP is a significant risk factor for all major CVD events including ACS, stroke, and mortality. However, these findings should be taken with caution due to the substantial heterogeneity and the possible publication bias.

## 1. Introduction

Community-acquired pneumonia (CAP) and cardiovascular disease (CVD) events are the leading causes of morbidity and mortality globally [1,2,3,4,5]. CAP is one of the most common reasons for adult hospital admissions. Over one million adults in the USA are hospitalized with pneumonia annually, and about 50,000 die from this disease [4,5,6]. Similarly, these diseases are associated with a significant social burden regarding health care resource utilization and social-economic cost [5].

There seems to exist a bidirectional relationship between pneumonia and CVD [7]. On one hand, CVDs such as coronary artery disease (CAD) and stroke increase the risk of hospitalization for pneumonia [7,8,9], but the opposite could also be true. That is, pneumonia may raise the risk of acute coronary syndrome (ACS)—myocardial infarction or unstable angina—stroke, heart failure, arrhythmias, and ev en death; acutely or even years after that [7,10,11].

However, the evidence linking CAP to cardiovascular complications is contradictory and not substantial. Most published studies included a single cohort without an adequate control group. Additionally, only two meta-analyses published in full text have included some of these studies [12,13]. Therefore, we aimed to systematically review the evidence on the role of CAP or respiratory tract infections as a risk factor for cardiovascular disease (CVD) complications.

## 2. Materials and Methods

We conducted this systematic review following the recommendations of the Cochrane Handbook for Systematic Reviews [14], PRISMA [15], and the AMSTAR 2 guidelines [16]. We previously registered the protocol in PROSPERO (CRD42022352910). We provide the PRISMA checklist in Figure 1. We searched for observational (cohort, case-control, and cross-sectional) studies and reviews published up until 31 December 2022, in Medline (PubMed), Google Scholar, Scopus, ScienceDirect, EMBASE, and Web of Science. We combined different keywords, controlled vocabulary terms (e.g., MeSH and Emtree terms), and free terms, according to the PECO strategy (population: “adults”; exposure: “pneumonia” OR “lower respiratory tract infection”; comparator: “no pneumonia” OR “no lower respiratory tract infection”; outcome: “acute coronary syndrome” OR “myocardial infarction” OR “unstable angina” OR “stroke” OR “mortality” OR “heart failure” OR “cardiac arrhythmia”) (Appendix A). Searches were not limited by date or language. We included articles in full text and excluded case reports, case series, studies not available in full text, duplicated publications, and studies with patients aged <18 years. Two independent reviewers examined the articles, and a third researcher resolved discrepancies. References from the retrieved papers were screened for additional articles.

The articles found were analyzed using the terms of the PECO strategy and the inclusion and exclusion criteria. In addition, relevant data from each paper were extracted and recorded in a spreadsheet: the name of authors, year and country of publication, type of study, number of patients, number of events, the measure of association, and adjusted confounders.

In the meta-analysis, we pooled the adjusted odds ratios (OR), risk ratios (RR), or hazard ratios (HR) with 95% confidence intervals (95% CI) using the generic inverse variance method. Forest plots represented the quantitative synthesis. Heterogeneity among studies was assessed with Cochran’s Q test and Higgins I^2^ statistic. Heterogeneity was significant (*p*-value < 0.05, I^2^ statistics >40%), then we used a random effects model. We carried out sensitivity and subgroup analyses. The risk of bias was assessed using the Newcastle–Ottawa Scale (NOS) tool [17] and publication bias was examined using a funnel plot.

We defined “all CVD events” as a composite of first ACS or first stroke event, new or worsening episodes of heart failure, arrhythmia (atrial fibrillation or paroxysmal supraventricular tachycardia), and death. Incident CVD event was defined as a composite of ACS, stroke, and fatal CHD [18]. For studies reporting OR or RR stratified into different subgroups, we considered each subgroup analysis as a separate study.

## 3. Results

We collected a total of 12,796 in the primary and 31 in the secondary search. After eliminating duplicates, 83 publications were evaluated for their titles and abstracts. Subsequently, 40 articles were analyzed in full text, of which 13 papers (nine cohort and four case-control studies) were selected for qualitative and quantitative assessment (Figure 1). We only included full-text articles that reported the adjusted association measures—OR, RR, or HR—and a control group. The lack of a proper control group was the leading cause for the exclusion of most studies [19,20,21,22,23,24,25,26,27,28,29,30,31,32,33,34,35,36,37,38,39,40,41,42,43,44,45,46,47,48,49,50,51,52,53,54,55,56,57,58,59,60,61,62,63,64,65,66,67] (Appendix A).

This study includes 276,109 participants, 18,298 first ACS events, 12,421 first stroke events, 119 arrhythmic events, 75 episodes of new onset or worsening HF, 3379 deaths, and 218 “incident CVD events” (Table 1). CAP increases the odds of ACS (OR 3.02; 95% CI 1.88–4.86), stroke (OR 2.88; 95% CI 2.09–3.96), mortality (OR 3.22; 95% CI 2.42–4.27), and all CVD events (OR 3.37; 95% CI 2.51–4.53). However, heterogeneity was significant (I^2^ = 97%, *p* < 0.001). The sensitivity analysis—with outliers excluded—did not significantly affect the overall estimate. In the subgroup analysis, we found statistically significant differences according to the continent of origin of the study (I^2^ = 78.2%, *p* < 0.10), the length of follow-up (I^2^ = 89.4%, *p* < 0.10), and the sample size (I^2^ = 75.1%, *p* < 0.10). Conversely, we did not find statistically significant differences between subgroups according to the study design (I^2^ = 0%, *p* < 0.93) (Figure 2A–G). Due to the limited data available, it was impossible to perform subgroup analysis according to other variables (i.e., the participants’ sex, the interval from exposition to the outcome, or cardiovascular risk levels). Similarly, we did not perform meta-regression analyses due to the limited number of studies.

All of the studies included had a low risk of bias (Table 2). However, the funnel plot suggested publication bias (Figure 3). 

## 4. Discussion

This systematic review and meta-analysis shows that CAP significantly increases the odds of developing ACS (OR 3.02; 95% CI 1.88–4.86), stroke (OR 2.88; 95% CI 2.09–3.96), mortality (OR 3.22; 95% CI 2.42–4.27), and all CVD events—a composite of first ACS event, first stroke event, new or worsening episodes of heart failure, arrhythmia (atrial fibrillation or paroxysmal supraventricular tachycardia), and death (OR 3.37; 95% CI 2.51–4.53) (Figure 2A–E). These findings are in agreement with other primary studies [7,68,69,70,71,72,73,74,75,76,77,78] and three meta-analyses [12,13,79].

Corrales-Medina et al. performed a meta-analysis to determine the incidence of major cardiac complications in CAP patients. They searched Medline, Scopus, and EMBASE for observational studies of adults with CAP reporting the following: overall cardiac complications, incident HF, ACS, or incident cardiac arrhythmias occurring within 30 days of CAP diagnosis. They found 25 articles that met the eligibility and minimum quality criteria. Seventeen articles (68%) reported cohorts of CAP inpatients. In this group, the pooled incidence rates for overall cardiac complications (six cohorts, 2119 patients), incident HF (eights cohorts, 4215 patients), ACS (six cohorts, 2657 patients), and incident cardiac arrhythmias (six cohorts, 2596 patients) were 17.7% (95% CI 13.9–22.2, 14.1% (95% CI 9.3–20.6), 5.3% (95% CI 3.2–8.6), and 4.7% (95% CI 2.4–8.9), respectively. One article reported cardiac complications in CAP outpatients, four in low-risk (not severely ill) inpatients, and three in high-risk inpatients. The incidences for all outcomes except the overall cardiac complications were lower in the two former groups and higher in the latter. One additional study reported on CAP outpatients and low-risk inpatients without discriminating between these groups. Twelve studies (48%) asserted the evaluation of cardiac complications in their methods, but only six (24%) defined them. Only three studies, all examining ACS, carried out risk factor analysis for these events. No study analyzed the association between cardiac complications and other medical complications or their impact on other CAP outcomes. Nevertheless, the authors concluded that major cardiac complications occur in a substantial proportion of patients with CAP [12].

Tralhão et al. undertook a meta-analysis to report the incidence of overall complications, ACS, new or worsening HF, new or worsening arrhythmias, and acute stroke as well as short-term mortality outcomes. In addition, they reviewed the interplay between the two conditions (pneumonia and CVD complications). They included 39 observational studies involving 92,188 patients, divided by setting (inpatients versus outpatients) and clinical severity (low risk versus high risk). They reported that the overall cardiac complications occurred in 13.9%, ACS in 4.5%, HF in 9.2%, arrhythmias in 7.2%, and stroke in 0.71% of the pooled inpatients. Furthermore, the meta-regression analysis suggested that overall and individual cardiac complication incidence decreased. After adjusting for confounders, cardiovascular events after CAP independently increased the risk for short-term mortality (range of OR: 1.39–5.49). The authors’ findings highlighted the need for effective, large, trial-based, preventive, and therapeutic interventions in this patient population [13].

Baskaran et al. performed a meta-analysis, searching for observational studies to summarize the literature on the incidence of ACS in adults with CAP. They looked for Medline and Embase and reported that 103 studies met the inclusion criteria. The authors included 26 studies (n = 66,347 patients), most of them of good quality. This meta-analysis showed that the pooled incidence of ACS in-hospital and 30 days after CAP was 3.2% (95% CI 2.4–4.0%; n = 17 studies) and 3.5% (95% CI 2.8–4.2%; n = 25 studies), respectively. Sensitivity analysis excluding studies with selected cohorts (elderly, predominantly male, or ICU admissions only) showed a higher pooled incidence for in-hospital ACS (4%, 95% CI 2.7–5.3%, n = 13 studies) but unchanged for 30-day incidence. These researchers concluded that their meta-analysis showed a small but significant risk of ACS in patients with CAP. This study was published in 2020 only in abstract form [79].

It is worth noting that most of the primary studies published to date and included in these three previous meta-analyses [12,13,79] had significant limitations. For example, most of them did not have an adequate control group or did not control for potential confounders (Appendix A), all of which may affect the real effect of the exposition or intervention [80,81,82,83]; therefore, we excluded more than 50 of these studies in our systematic review [19,20,21,22,23,24,25,26,27,28,29,30,31,32,33,34,35,36,37,38,39,40,41,42,43,44,45,46,47,48,49,50,51,52,53,54,55,56,57,58,59,60,61,62,63,64,65,66,67]. Furthermore, the meta-analyses by Corrales-Medina et al. [12], Tralhão et al. [13], and Baskaran et al. [79] are “meta-analyses of proportion”. A “proportional meta-analysis” differs significantly in its methodology from a “traditional meta-analysis”. We address these aspects later. Furthermore, the study by Baskaran et al. was published only in abstract form [79], so it is impossible to know what studies were included. Despite the limitations of the three meta-analyses described previously, their conclusions were concordant with our study [12,13,79].

The pathophysiologic mechanisms by which pneumonia can trigger CVD are probably diverse and involve several ways: (1) systemic and coronary artery inflammation increases cardiovascular risk; (2) infection and inflammation promote platelet activation and thrombosis; (3) changes in nitric oxide (NO) synthase and cyclooxygenase (COX) lead to endothelial dysfunction; (4) pneumonia impairs myocardial contractility, oxygen demand, and delivery; and additionally (5), the microorganisms may have a direct effect on cardiovascular risk [84,85,86,87,88,89]. However, most of the mechanisms above-mentioned were also observed in ischemic heart disease patients, particularly those with ACS. Thus, these findings may also affect pneumonia development after ACS, reflecting a reverse or bidirectional association [8,23,84,88].

To the best of our knowledge, this is the first “non-proportional” meta-analysis on CAP and CVD that reports ORs instead of proportions as a measure of effect size. A proportional meta-analysis is a data synthesis method that allows one to calculate a pooled, overall proportion from several individual proportions for a certain event instead of estimating an effect size, as conducted in a “traditional meta-analysis” [90,91]. That is, a proportional meta-analysis can include dichotomous data reported as a percentage. Proportional meta-analysis is encouraged when conducting systematic reviews of prevalence, incidence, and maybe, interventions and therapies where appropriate [92]. However, for dichotomous outcomes, the Cochrane Handbook does not recommend using proportional meta-analysis [14].

Furthermore, the appropriateness of conducting a proportional meta-analysis is controversial, as the individual studies contributing to such a meta-analysis commonly have been conducted in different contexts. These studies’ prevalence and cumulative incidence estimates reflect unique population characteristics [93]. This methodology raises concerns when proportional meta-analysis assumes homogeneity, and an average estimate across different populations may be of little clinical use [92,94]. However, Corrales-Medina et al. [12] and Tralhão et al. [13]. reported that they used stratification (mainly according to treatment setting and clinical severity) to minimize the influence heterogeneity in estimating the effect sizes.

This work has some limitations. (1) Although we performed subgroup analysis according to the continent of origin, the type of study design, the sample size, and the length of follow-up, we could not perform subgroup analyses according to other important variables such as age, sex, treatment setting, clinical severity, and timing after CAP because of the lack of data. (2) It is possible that a meta-regression analysis could further explain the origin of the heterogeneity, although we did not perform this analysis due to limited data. (3) We cannot rule out a possible publication bias against negative studies that did not find a significant association between CAP and CVD complications. (4) The fact that the different continent of origin explains the heterogeneity found could reflect the different forms of diagnosis, treatment, and prognosis, both for pneumonia and cardiovascular complications in each of these countries.

Overall, if the heterogeneity is high, the conclusions of the meta-analysis may be less generalizable. Nevertheless, this does not necessarily mean that the results are incorrect or do not have clinical importance. Instead, it is crucial to interpret the meta-analysis results considering the heterogeneity and possible explanations for the differences between the studies included [44,95,96,97]. In addition, there is always clinical and methodological diversity in a meta-analysis, so statistical heterogeneity is inevitable. Since systematic reviews bring together studies that are diverse both clinically and methodologically, heterogeneity in their results is to be expected [95,96,97]. In our study, heterogeneity was significant in most of the outcomes. However, we addressed this heterogeneity by implementing the recommendations of the Cochrane Handbook: (1) We verified the data correctness; (2) we explored the potential causes of heterogeneity; (3) we performed a meta-analysis with a random effects model, and (4) we performed sensitivity analyses [14].

We highlight some of the strengths of our work: (1) Our search strategy was thorough and complete; (2) we included the odds ratios instead of proportion as the effect measure; consequently, this is the first “traditional” meta-analysis, instead of proportional meta-analysis on this topic; (3) we included primary studies that specifically examined the association between pneumonia and CVD complications; (4) we excluded studies that reported a single cohort without a control group; and (5) we only included studies that reported the adjusted effect sizes. Therefore, our results are more robust than any other meta-analysis reported before.

## 5. Conclusions

In conclusion, our study shows that pneumonia should be considered as a new risk factor for cardiovascular complications. Furthermore, our findings support the hypothesis that inflammation triggered by acute and chronic infections such as pneumonia is crucial in the pathogenesis of atherosclerosis and cardiovascular complications. However, this conclusion should be taken with caution due to the limitations of our study.

## Figures and Tables

**Figure 1 jcm-12-02577-f001:**
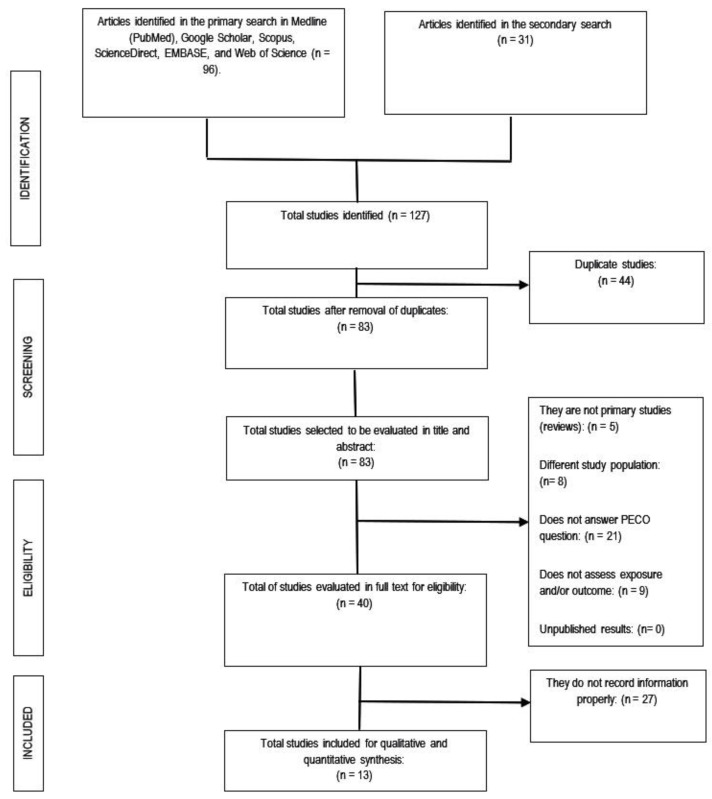
Flowchart of the selection process of the primary studies included.

**Figure 2 jcm-12-02577-f002:**
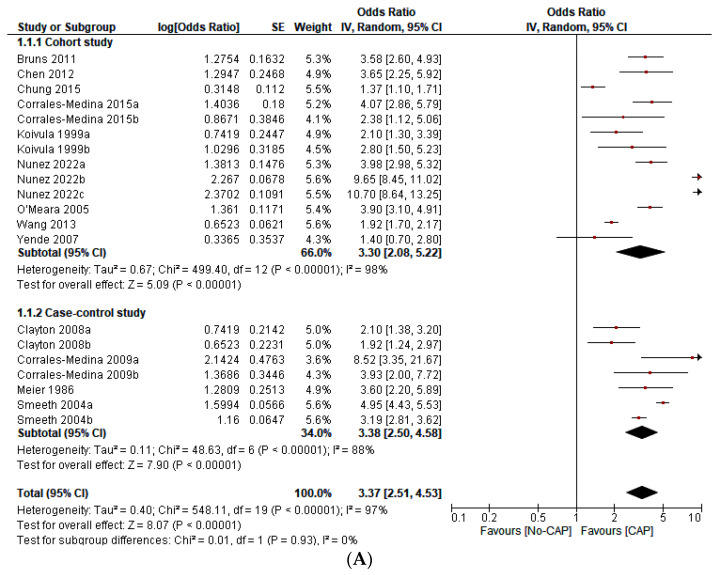
(**A**) Forest plot on the effect of CAP on all CVD events (a composite of first ACS or first stroke event, new or worsening episodes of heart failure, arrhythmia, and death) according to the type of study design [1,2,3,4,5,6,7,8,9,10,13,15]. (**B**) Forest plot of the effect of CAP on all CVD events (a composite of the first ACS or first stroke event, new or worsening episodes of heart failure, arrhythmia, and death) according to the continent of origin of the study [4,8,11,12,13]. (**C**) Forest plot of the effect of CAP on all CVD events (a composite of first ACS or first stroke event, new or worsening episodes of heart failure, arrhythmia, and death) according to the length of follow-up in years [1,2,3,4,5,6,7,8,9,10,11,12,13]. (**D**) Forest plot of the effect of CAP on all CVD events (a composite of first ACS or first stroke event, new or worsening episodes of heart failure, arrhythmia, and death), according to the sample size in quartiles [1,2,3,4,5,6,7,8,9,10,11,12,13]. (**E**) Forest plot of the effect of CAP on the first ACS event [1,2,3,4,5,9,12]. (**F**) Forest plot of the effect of CAP on the first stroke event [2,3,10]. (**G**) Forest plot of the effect of CAP on mortality [6,7,8,12,13].

**Figure 3 jcm-12-02577-f003:**
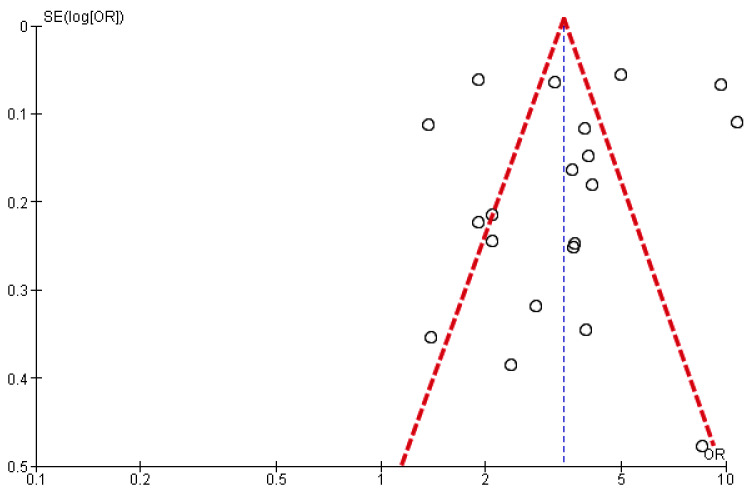
Funnel plot of the effect of CAP on all CVD events (a composite of the first ACS or first stroke event, new, or worsening episodes of heart failure, arrhythmia, and death).

**Table 1 jcm-12-02577-t001:** General characteristics of the studies included.

Study, Year (Region)	Participants, Study Design, Sample Characteristics	Exposition	Outcome	Adjustment Factors	OR/RR/HR (95% CI)
Meier [1], UK, 1986	N = 9571. Cases (MI) 1922, controls (no MI) 7649. Two separate analyses: case-control and case-crossover study. Follow-up: 3 y. Both sexes. Age ≤75 years. Deaths among cases 285. Deaths among controls NR.	Acute RTI	First MI	Smoking and BMI	AOR = for first-time MI at 1–5, 6–10, 11–15, or 16–30 days after ARTI were 3.6 (2.2–5.7), 2.3 (1.3–4.2), 1.8 (1.0–3.3), and 1.0 (0.7–1.6). RR = 2.7 (1.6–4.7) for MI at 10 days after ARTI.
Smeeth [2], UK; 2004a	N = 65,746. First MI 53709. ARTI 20,921. ARTI and first MI 3254. Mean follow-up 5.6 y. Both sexes. Median age at MI 72.3 y. Case-series method.	Acute RTI	First MI	Age	AOR = for first MI at 1–3, 4–7, 8–14, 15–28, 29–91 days since ARTI were 4.95 (4.43–5.53), 3.20 (2.84–3.60), 2.81 (2.54–3.09), 1.95 (1.79–2.12), 1.95 (1.79–2.12), respectively.
Smeeth [2], UK; 2004b	N = 66,637 patients. First stroke 50,766. ARTI 22,400. ARTI and first stroke 3060. Mean follow-up 5.3 y. Both sexes. Median age at stroke 78.3 y. Case-series method.	Acute RTI	First stroke	Age	AOR = for first stroke at 1–3, 4–7, 8–14, 15–28, 29–91 days since ARTI were 3.19 (2.81–3.62), 2.34 (2.05–2.66), 2.09 (1.89–2.32), 1.68 (1.54–1.82), 1.33 (1.26–1.40), respectively.
Clayton [3], UK, 2008a	Cases (MI) 11,155. Controls (no MI) 11,155. Mean follow-up 1 y. Both sexes. Median age at MI 79 ± 14 y. Case-control study.	Acute RTI	First MI	Angina, smoking, DM, HT, PVD, family history of CAD, hyperlipidemia, previous stroke.	AOR = for first MI 2.10 (1.38–3.21), 1.93 (1.42–2.63), 1.16 (0.92–1.47), 1.08 (0.94–1.23), during the 1–7, 8–28, 29–91, 92–365 days following infection, respectively.
Clayton [3], UK, 2008b	Cases (stroke) 9208. Controls (no stroke) 9208. Mean follow-up 1 y. Both sexes. Median age at stroke 74 ± 13 y. Case-control study.	Acute RTI	First stroke	Smoking, DM, HT, PVD, previous MI, UTI	AOR = for stroke 1.92 (1.24–2.97), 1.76 (1.27–2.45), 1.09 (0.88–1.36), 1.08 (0.94–1.24), during the 1–7, 8–28, 29–91, 92–365 days following infection, respectively.
Nuñez-Delgado [4], Peru, 2022a	N = 693 (CAP 231, no CAP 462). Ambispective cohort. Follow-up: 2 y. CAP and MI 107, no CAP and MI 0. Both sexes. Age >30 years. Mean age 64.1 ± 13.7 years.	CAP	ACS (MI)	Smoking, HT, DM, hypercholesterolemia	ARR = 3.98 (2.98–5.33) for ACS.
Nuñez-Delgado [4], Peru, 2022b	N = 693 (CAP 231, no CAP 462). Ambispective cohort. Follow-up: 2 y. CAP and HF 75, no CAP and HF 0. Both sexes. Age >30 years. Mean age 64.1 ± 13.7 years.	CAP	HF	Smoking, HT, DM, hypercholesterolemia	ARR = 9.65 (8.45–11.0) for HF.
Nuñez-Delgado [4], Peru, 2022c	N = 693 (CAP 231, no CAP 462). Ambispective cohort. Follow-up: 2 years. CAP and arrhythmia 119, no CAP and arrhythmia 0. Both sexes. Age > 30 years. Mean age 64.1 ± 13.7 years.	CAP	Arrhythmias (AF, PSVT)	Smoking, HT, DM, hypercholesterolemia	ARR = 10.7 (8.64–13.2) for arrhythmias.
Wang [5], Taiwan, 2013	CAP 20,111, no CAP 80,444. Prospective cohort study. CAP and ACS 1044, no CAP and ACS 332. Both sexes. Age ≥20 years. Follow-up 14 y.	CAP	First episode of ACS.	Age, sex, comorbidities (HT, DM, dyslipidemia, COPD).	ARR 1.92 (1.70–2.17) for ACS. ARR 3.90 (2.46–6.18) within 3 months; ARR = 2.43 (1.75–3.38) within 1 year, ARR 1.74 (1.51–2.00) >1 year. AHR = 1.47 (1.24–1.73) for ACS in the following 14 years. AHR = 1.18 (1.02–1.37) for ACS in males.
Koivula [6], Finland, 1999a	N = 4167. CAP 122. Follow-up 9.2 y. Prospective observational (cohort) study. No CAP 4045. Both sexes. Age ≥60 y. Deaths 1979. Mean follow-up 9.2 y.	CAP	Total mortality, cardiovascular mortality	Age, sex, and multiple comorbidities.	ARR 2.1 (1.3–3.4) for pneumonia-related mortality. ARR 1.5 (1.2–1.9) for total mortality. ARR 1.4 (1.0–1.9) for cardiovascular mortality.
Koivula [6], Finland, 1999b	N = 4167. PCAP 53. Follow-up 9.2 y. Prospective observational (cohort) study. No CAP 4045. Both sexes. Age ≥60 y. Deaths 1979. Mean follow-up 9.2 y.	PCAP	Total mortality, cardiovascular mortality	Age, sex, and multiple comorbidities.	RR 2.8 (1.5–5.3) for pneumonia-related mortality. ARR 1.6 (1.1–2.2) for total mortality. ARR 1.6 (1.0–2.4) for cardiovascular mortality.
Bruns [7], Netherlands, 2011	N = 712. Patients discharged from hospital after an episode of CAP 356. Death in CAP 187, death in no CAP 85. Follow-up: 7 y. Both sexes. Age ≥18 y. Mean age of the CAP patients. Follow-up 7 y. 66.0 ± 16.1 years. Prospective cohort study.	CAP	Mortality rate	Age, sex, PSI	AOR 3.58 (2.60–4.94) for long-term mortality rate.
Yende [8], USA, 2007	N = 3075, 106 subjects hospitalized for CAP. Follow-up: 5.2 y. Prospective cohort study. Deaths: 361. Both sexes. Age 70–79 y. Mean age 73.6 ± 2.9 y.	CAP	Mortality	Age, sex, race, site, smoking, DM, CHD, eGFR, FEV1, albuminemia, cognitive function, functional status, TNF, IL-6.	AOR 1.4 (0.7–3.0) for mortality at 0–30 days. AOR 3.5 (1.5–8.1) for mortality at 31–365 days. AOR 5.6 (2.8–11.2) for mortality at >365 days.
Chung [9], Taiwan, 2015.	N = 12,152 newly diagnosed MP. No MP 48,600 individuals. Nationwide longitudinal cohort study. Follow-up up to >12 months. Both sexes. ACS and MP 350. ACS and no MP 106.	MP	New ACS (unstable angina and MI).	Sex, age, comorbidities and follow-up time.	AHR 1.37(1.10–1.70) for ACS. AHR 1.49 (1.06–2.08) for ACS in females. AHR 1.29 (0.97–1.71) for ACS in males. AHR 1.48 (1.01–2.16) for ACS in ≤64 y. AHR 1.34 (1.02–1.74) for ACS in >65 y.
Chen [10], Taiwan, 2012.	Hospitalized patients. PCAP 745, no PCAP 1490. Cohort study. PCAP and stroke 80, no PCAP and stroke 73. Follow-up: 2 y. Both sexes. Age > 18 y. In both cohorts >60% were ≥65 y.	PCAP	Stroke	Patient characteristics, comorbidities, geographic region, urbanization, level of residence, and socioeconomic status.	AHR 3.65 (2.25–5.90) for stroke in the first year. AHR 0.91 (0.53–1.59) for stroke in the second year. AHR 5.00 (1.78–14.07) for stroke in the first year in those with comorbidities. AHR 3.23(1.86–5.62) for stroke in the first year in those without comorbidities.
Corrales-Medina [11], USA, 2015a	Community-based prospective cohort. CHS cohort. Age ≥ 65 years, CAP 591, no CAP 1182. Both sexes. CVD * events 173. Follow-up up to 10 y.	CAP	Incident CVD (MI, stroke, and fatal CHD)	Age, sex, race, HT, DM, total cholesterol, HDL, LDL, smoking, alcohol abuse, AF, CKD, CRP, CVD, FEV1, daily living activities, modified MMS score.	AHR 4.07 (2.86–5.27) for CVD at 0–30 d. AHR 2.94 (2.18–3.70) for CVD at 31–90 d. AHR 2.10 (1.59–2.60) for CVD at 91 d-1 y. AHR 1.86 (1.18–2.55) for CVD at 9–10 y.
Corrales-Medina [11], USA, 2015b	Community-based prospective cohort. ARIC cohort. Age 45–64 years, CAP 680, no CAP 1360. Both sexes. CVD * events 45. Follow-up up to 10 y.	CAP	Incident CVD (MI, stroke, and fatal CHD)	Age, sex, race, HT, DM, total cholesterol, HDL, LDL, smoking, alcohol abuse, AF, CKD, Q waves in ECG, PAD, FEV1	AHR 2.38 (1.12–3.63) for CVD at 0–30 d. AHR 2.40 (1.23–3.47) for CVD at 31–90 d. AHR 2.19 (1.20–3.19) for CVD at 91 d-1 y. AHR 1.88 (1.10–2.66) for CVD at 9–10 y.
Corrales-Medina [12], USA, 2009a	Case-control study. CAP patients (144 S. pneumoniae, 62 H. influenzae) 206. Controls 395. ACS: 22 cases among CAP patients and 6 among 395 controls. Both sexes. Follow-up 475 d.	PCAP or HCAP	ACS	CHD equivalent (CHD, or cerebrovascular disease, or PVD, HF, ≥2 coronary risk factors (DM, HT, dyslipidemia, smoking, family history of CHD).	AOR 8.52 (3.35–22.23) for ACS.
Corrales-Medina [12], USA, 2009b	Case-control study. CAP patients (144 S. pneumoniae, 62 H. influenzae) 206. Controls 395. Thirty-day mortality: 26 cases among CAP patients and 14 among 395 controls. Both sexes. Follow-up 475 d.	PCAP or HCAP	Thirty-day mortality	CHD equivalent (CHD, or cerebrovascular disease, or PVD, HF, ≥2 coronary risk factors (DM, HT, dyslipidemia, smoking, family history of CHD).	AOR 3.93 (2.00–22.7.71) for 30-day mortality.
O’Meara [13], USA, 2005	CHS. N = 5888 men and women aged ≥65. CAP 582. No CAP: 5306. Median follow-up 10.7 years. Prospective cohort.	CAP	Total mortality	* Age, sex, and race. ** Age, sex, and race, Baseline history of CVD, DM, smoking, and measures of lung, physical, and cognitive function.	* ARR 4.9 (4.1–6.0) for total mortality during the first year after hospitalization. * ARR 2.6 (2.2–3.1) for total mortality after the first year after hospitalization. ** ARR 3.9 (3.1–4.8) for total mortality during the first year after hospitalization. ** ARR 2.0 (1.6–2.4) for total mortality after the first year after hospitalization.

CVD: cardiovascular disease, PVD: peripheral vascular disease, UTI: urinary tract infection, RTI: respiratory tract infection, BMI: body mass index, PSI: pneumonia severity index, CXR: chest X ray; DM: diabetes mellitus, CHD coronary heart disease, eGFR: estimated glomerular filtration rate (<60 mL/min), FEV1: forced expiratory volume in 1 s, TNF: circulating concentrations of tumor necrosis factor, IL-6: circulating concentrations of interleukin-6, PCAP: CAP caused by pneumococcus, HCAP: CAP caused by H. influenzae, COPD: chronic obstructive pulmonary disease, history of pneumococcal infection, TIA: transient ischemic attack, CLD: chronic liver disease, CKD: chronic kidney disease, MP: mycoplasma pneumonia, CVE: cardiovascular event, CRP: C-reactive protein, AF: atrial fibrillation, CVD * events: composite of MI + stroke + fatal CHD; AF: atrial fibrillation, PSVT: parodistic supraventricular tachycardia, MMS: mini mental status, ARIC: Atherosclerosis Risk in Communities study, CHS Cardiovascular Health Study, NR: not reported.

**Table 2 jcm-12-02577-t002:** Bias assessment of the included primary studies.

Author	Study Design	Tool	Selection	Comparability	Outcome	Total	Conclusion
Meier [1], UK, 1986	CC	NOS	***	**	**	7	Low risk
Smeeth [2], UK; 2004	CC	NOS	***	**	**	7	Low risk
Clayton [3], UK, 2008	CC	NOS	***	**	**	7	Low risk
Nuñez-Delgado [4], Peru, 2022	CS	NOS	***	**	***	8	Low risk
Wang [5], Taiwan, 2013	CC	NOS	***	**	**	7	Low risk
Koivula [6], Finland, 1999	CS	NOS	***	**	***	8	Low risk
Bruns [7], Netherlands, 2011	CS	NOS	***	**	**	7	Low risk
Yende [8], USA, 2007	CC	NOS	***	**	**	7	Low risk
Chung [9], Taiwan, 2015.	CS	NOS	***	**	***	8	Low risk
Chen [10], Taiwan, 2012.	CS	NOS	***	**	**	7	Low risk
Corrales-Medina [11], USA, 2015	CS	NOS	****	**	***	9	Low risk
Corrales-Medina [12], USA, 2009	CC	NOS	***	**	***	8	Low risk
O’Meara [13], 2005, USA	CS	NOS	***	**	**	7	Low risk

CC: case control study, CS: cohort study. Note: An asterisk (*) represents a star in each domain of the Newcastle–Ottawa Scale (NOS) tool.

## Data Availability

The protocol is available at https://www.crd.york.ac.uk/prospero/export_details_pdf.php (accessed on 31 January 2022).

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
