# Peer review of "Acute Coronary Syndrome, Stroke, and Mortality after Community-Acquired Pneumonia: Systematic Review and Meta-Analysis"

_jcm, 2023, doi:10.3390/jcm12072577_

Round 1

Reviewer 1 Report

The author has written a manuscript titled "Acute coronary syndrome, stroke, and mortality after community-acquired pneumonia: systematic review and meta-analysis". 

Major comment: 

1: High hetroginity with all primary and secondary outcomes. Meaning result cant be generalised.

2: A lot of research ( SRMA) on the same topic has been done already with almost the same findings. What is new in this paper compared to all previous meta-analyses?

3: I would recommend author to wait for some RCTs to be added as with so high 98,99% hetroginity in findings are not so predictive.

4: Author must have also performed regression and also subgroup analysis based on sample size, gender, followup, high vs low risk cvd group

Author Response

Reviewer 1

The author has written a manuscript titled “Acute coronary syndrome, stroke, and mortality after community-acquired pneumonia: systematic review and meta-analysis”. 

Major comment: 

1: High heterogeneity with all primary and secondary outcomes. Meaning result can’t be generalized.

Overall, if heterogeneity is high, the conclusions of the meta-analysis may be less generalizable. However, this does not necessarily mean that the results are incorrect or do not have clinical importance. Instead, it is crucial to interpret the meta-analysis results considering heterogeneity and possible explanations for the differences between the studies included. In addition, there is always clinical and methodological diversity in a meta-analysis, then statistical heterogeneity is inevitable. Since systematic reviews bring together studies that are diverse both clinically and methodologically, heterogeneity in their results is to be expected. Although it is true that in our study, in most of the outcomes, the heterogeneity was significant, however, we addressed this heterogeneity by implementing the recommendations of the Cochrane Handbook for Systematic Reviews of Interventions: 1) we verified the correctness of the data, 2) we explored the potential causes of heterogeneity, 3) we performed a meta-analysis with a random effects model, and 4) we sequentially excluded studies with extreme values (sensitivity analysis). We added a paragraph in Discusion explaining this issue.

References

  1. Higgins JPT, Green S (editors). Cochrane Handbook for Systematic Reviews of Interventions. Version 5.1.0 [updated March 2011]. The Cochrane Collaboration, 2011. Available from: https://handbook-5-1.cochrane.org/
  2. Higgins JP, Thompson SG, Deeks JJ, Altman DG. Measuring inconsistency in meta-analyses. BMJ. 2003 Sep 6;327(7414):557-60. doi: 10.1136/bmj.327.7414.557.
  3. Ioannidis JP, Patsopoulos NA, Evangelou E. Heterogeneity in meta-analyses of genome-wide association investigations. PLoS One. 2007;2(9):e841. doi: 10.1371/journal.pone.0000841.
  4. Thompson SG, Sharp SJ. Explaining heterogeneity in meta-analysis: a comparison of methods. Stat Med. 1999 Oct 30;18(20):2693–708. doi: 10.1002/(sici)1097-0258(19991030)18:20<2693::aid-sim235>3.0.co;2-v

2: A lot of research (SRMA) on the same topic has been done already with almost the same findings. What is new in this paper compared to all previous meta-analyses?

Although indeed, there are already some—not a lot—published systematic reviews that have investigated a similar research question, we highlight the main strengths of our study: 1) our research gathers more participants and CVD events than any other SRMA, 2) we included odds ratios instead of proportions as the effect measure; consequently, this is the first “traditional” meta-analysis, instead of proportional meta-analysis, on this topic, 3) we included studies that specifically examined the association between pneumonia and CVD complications, 4) we excluded studies that reported a single cohort without a control group, and 5) we only included studies that reported adjusted effect sizes. Therefore, our results are more robust than any other SRMA published before.

We expanded a paragraph in Discusion explaining this issue.

3: I would recommend author to wait for some RCTs to be added as with so high 98,99% heterogeneity in findings are not so predictive.

A randomized controlled trial (RCT) is a type of scientific experiment used to evaluate the effectiveness of an intervention, in comparison to a control group that receives either a placebo or standard treatment. The RCT provide the best evidence and is considered the gold standard for evaluating the efficacy and safety of interventions because it can establish a cause-and-effect relationship between the intervention and the outcome. However (1-5), RCTs must follow certain ethical and legal principles to protect participants. The fundamental ethical principle in research with human beings is respect for human dignity, which implies that people must be treated as autonomous subjects and not as research objects. Consequently, RCT must be designed in a way that minimizes the risks to participants, and the potential benefits must outweigh the risks. In the case of exposing a group of people to a serious disease such as pneumonia, the risks to the health and life of the participants would be too high and would not justify the potential benefits. Therefore, we consider that what the reviewer comments is not feasible, since it is highly unlikely that any ethics committee will authorize an RCT on this topic since deliberately exposing participants to unnecessary harm, such as a disease for the purpose of research is unethical and prohibited.

References

  1. Ioannidis JP, Evans SJ, Gøtzsche PC, et al. Better reporting of harms in randomized trials: an extension of the CONSORT statement. Ann Intern Med. 2004;141(10):781-788. doi:10.7326/0003-4819-141-10-200411160-00009
  2. Council for International Organizations of Medical Sciences (CIOMS). International Ethical Guidelines for Health-related Research Involving Humans. Geneva: CIOMS; 2016. Available at: https://cioms.ch/wp-content/uploads/2017/01/WEB-CIOMS-EthicalGuidelines.pdf
  3. World Health Organization (WHO). Standards and Operational Guidance for Ethics Review of Health-related Research with Human Participants. 2nd ed. Geneva: WHO; 2011. Available at: https://www.who.int/ethics/publications/research_standards_9789241502948/en/
  4. Nuffield Council on Bioethics. Medical profiling and online medicine: the ethical implications. London: Nuffield Council on Bioethics; 2010. Available at: https://www.nuffieldbioethics.org/wp-content/uploads/2014/07/Medical-profiling-and-online-medicine-the-ethical-implications.pdf
  5. Emanuel EJ, Wendler D, Killen J, Grady C. What makes clinical research ethical? JAMA. 2000;283(20):2701-2711. doi:10.1001/jama.283.20.2701
  6. The Belmont Report: Ethical Principles and Guidelines for the Protection of Human Subjects of Research. The National Commission for the Protection of Human Subjects of Biomedical and Behavioral Research; 1979. Available at: https://www.hhs.gov/ohrp/regulations-and-policy/belmont-report/read-the-belmont-report/index.html

4: Author must have also performed regression and also subgroup analysis based on sample size, gender, follow-up, high vs low risk cvd group.

We have performed subgroup analysis based on sample size and length of follow-up. We added these subgroup analyses to the results.

However, since almost all included studies did not present their results separately according to gender or cardiovascular risk levels, performing a subgroup analysis for these variables was impossible.

The authors appreciate these valuable comments that will enrich the methodological quality of this work.

Reviewer 2 Report

The authors of this manuscript performed a meta-analysis of 13 observational studies that have investigated the association between community-acquired pneumonia (CAP) and coronary syndrome (ACS), stroke, heart failure (HF), arrhythmias, or mortality. Overall, these studies included: 276,109 participants, 18,298 first ACS events, 12,421 first 17 stroke events, 119 arrhythmic events, 75 episodes of new onset or worsening heart failure, 3,379 deaths, and 18 218 incident CVD events.  CAP increased the odds of ACS (OR 3.02; 95% CI 1.88–4.86), stroke (OR 19 2.88; 95% CI 2.09–3.96), mortality (OR 3.22; 95% CI 2.42–4.27), and all cardiovascular disease (CVD) events (OR 3.37; 95% CI 20 2.51–4.53). Heterogeneity was significant (I2 = 97%, p < 0.001), and subgroup analysis found differences according to the continent of origin of the study (I² = 78.2%, p < 0.10). The authors concluded that CAP is a significant risk factor for all major CVD events, including ACS, stroke, and mortality. However, caution is warranted in interpretation of the data due to significant heterogeneity and publication bias.

This an important and clinically relevant work in this field. In my knowledge, the meta-analytic methodology is properly used and the main findings of the meta-analysis are intuitively correct. CAP represent a high-inflammatory state, both locally and systemically, with deleterious impact on atherosclerotic vascular disease throughout the body. I have only minor comments related to the study:

1.      There are at least 3 previous meta-analyses on the same subject. I advise the authors to evidence advantages of current meta-analyses compared with previous meta-analyses rather than discussing at length their findings, as they did in the discussion section. Thus, please emphasize what’s new with this meta-analysis?

2.      The significant heterogeneity for all outcomes analyzed is rather problematic. How can these findings be interpreted in face of such high heterogeneity? As stated, the meta-regression could not be performed. Tralhão and Póvoa managed to perform a meta-regression in their meta-analysis (reference 13). Can the authors offer some insights on how to solve this problem? At least. The limitation section could be expanded.

3.      The authors stated that publication bias favoring the negative studies were possible. Could it be that the authors used a restricted search strategy. Is it because of nature of the studies or because of search (and inclusion) strategy?

4.      In the discussion section, the authors emphasized the mutual relationship between CVD and susceptibility to CAP. Although this is true, in the setting of current meta-analysis, the impact of CAP on CVD and not vice versa should be the main focus.

Author Response

Reviewer 2

The authors of this manuscript performed a meta-analysis of 13 observational studies that have investigated the association between community-acquired pneumonia (CAP) and coronary syndrome (ACS), stroke, heart failure (HF), arrhythmias, or mortality. Overall, these studies included: 276,109 participants, 18,298 first ACS events, 12,421 first 17 stroke events, 119 arrhythmic events, 75 episodes of new onset or worsening heart failure, 3,379 deaths, and 18 218 incident CVD events.  CAP increased the odds of ACS (OR 3.02; 95% CI 1.88–4.86), stroke (OR 19 2.88; 95% CI 2.09–3.96), mortality (OR 3.22; 95% CI 2.42–4.27), and all cardiovascular disease (CVD) events (OR 3.37; 95% CI 20 2.51–4.53). Heterogeneity was significant (I2 = 97%, p < 0.001), and subgroup analysis found differences according to the continent of origin of the study (I² = 78.2%, p < 0.10). The authors concluded that CAP is a significant risk factor for all major CVD events, including ACS, stroke, and mortality. However, caution is warranted in interpretation of the data due to significant heterogeneity and publication bias.

This an important and clinically relevant work in this field. In my knowledge, the meta-analytic methodology is properly used and the main findings of the meta-analysis are intuitively correct. CAP represent a high-inflammatory state, both locally and systemically, with deleterious impact on atherosclerotic vascular disease throughout the body. I have only minor comments related to the study:

  1. There are at least 3 previous meta-analyses on the same subject. I advise the authors to evidence advantages of current meta-analyses compared with previous meta-analyses rather than discussing at length their findings, as they did in the discussion section. Thus, please emphasize what’s new with this meta-analysis?

Although indeed, there are already some published systematic reviews that have investigated a similar research question, we highlight the main strengths of our study: 1) our research gathers more participants and CVD events than any other SRMA, 2) we included odds ratios instead of proportions as the effect measure; consequently, this is the first “traditional” meta-analysis, instead of proportional meta-analysis on this topic 3) we included studies that specifically examined the association between pneumonia and CVD complications, 4) we excluded studies that reported a single cohort without a control group, and 5) we only included studies that reported adjusted effect sizes. Therefore, our results are more robust than any other SRMA published before. We addressed the pitfalls of proportional meta-analysis in the penultimate paragraph of our Discussion.

  1. The significant heterogeneity for all outcomes analyzed is rather problematic. How can these findings be interpreted in face of such high heterogeneity? As stated, the meta-regression could not be performed. Tralhão and Póvoa managed to perform a meta-regression in their meta-analysis (reference 13). Can the authors offer some insights on how to solve this problem? At least. The limitation section could be expanded.

Tralhão et al. performed a meta-regression in their meta-analysis because they carried out a “proportional metanalysis”, like the other two meta-analyses included in our Discussion. Besides, they included more studies—-but not more participants or events—than our study. Nonetheless, as we describe in Discusion, we excluded most of the studies included in the study of Tralhão et al. and Corrales-Medina et al. because these studies did not include a control group or not reported adjusted of cofounders. In our systematic review, we included more recent studies, all of which had a control group and adjustment of cofounders. Even further, in this revised version of our manuscript, we have performed subgroup analysis according to sample size and length of follow-up. In addition, we have expanded the Limitation section explaining this issue.

  1. The authors stated that publication bias favoring the negative studies were possible. Could it be that the authors used a restricted search strategy. Is it because of nature of the studies or because of search (and inclusion) strategy?

As we stated in Strengths, our search strategy (see Supplementary materials) was thorough and sensitive and included six databases. In fact, we identified much more studies than any other previous systematic review. Nonetheless, we excluded 53 studies (see Supplementary materials) because these studies did not include a control group or did not report adjusted of cofounders.

  1. In the discussion section, the authors emphasized the mutual relationship between CVD and susceptibility to CAP. Although this is true, in the setting of current meta-analysis, the impact of CAP on CVD and not vice versa should be the main focus.

We do not talk about this mutual or bidirectional relationship between pneumonia and CVD in Discussion. In fact, we only mention it in the second paragraph of the Introduction.

The authors appreciate these valuable comments that will enrich the methodological quality of this work.

Round 2

Reviewer 1 Report

Thanks for clarification